# Mechanochemical coupling and bi-phasic force-velocity dependence in the ultra-fast ring ATPase SpoIIIE

**Ninning Liu[1,2†‡], Gheorghe Chistol[1,3†§], Yuanbo Cui[1,4], Carlos Bustamante[1,2,3,4,5,6,7]\***

[1]Jason L. Choy Laboratory of Single Molecule Biophysics, University of California, Berkeley, Berkeley, United States; [2]Department of Molecular and Cell Biology, University of California, Berkeley, Berkeley, United States; [3]Department of Physics, University of California, Berkeley, Berkeley, United States; [4]California Institute for Quantitative Biosciences, University of California, Berkeley, Berkeley, United States; [5]Department of Chemistry and Howard Hughes Medical Institute, University of California, Berkeley, Berkeley, United States; [6]Physical Biosciences Division, Lawrence Berkeley National Laboratory, Berkeley, United States; [7]Kavli Energy NanoSciences Institute at the University of California, Berkeley and the Lawrence Berkeley National Laboratory, Berkeley, United States

**\*For correspondence:**
carlosb@berkeley.edu

[†]These authors contributed equally to this work

**Present address:** [‡]Wyss Institute, Boston, United States; [§]Department of Biological Chemistry and Molecular Pharmacology, Harvard Medical School, Boston, United States

**Competing interests:** The authors declare that no competing interests exist.

**Abstract** Multi-subunit ring-shaped ATPases are molecular motors that harness chemical free energy to perform vital mechanical tasks such as polypeptide translocation, DNA unwinding, and chromosome segregation. Previously we reported the intersubunit coordination and stepping behavior of the hexameric ring-shaped ATPase SpoIIIE (Liu et al., 2015). Here we use optical tweezers to characterize the motor's mechanochemistry. Analysis of the motor response to external force at various nucleotide concentrations identifies phosphate release as the likely force-generating step. Analysis of SpoIIIE pausing indicates that pauses are off-pathway events. Characterization of SpoIIIE slipping behavior reveals that individual motor subunits engage DNA upon ATP binding. Furthermore, we find that SpoIIIE's velocity exhibits an intriguing bi-phasic dependence on force. We hypothesize that this behavior is an adaptation of ultra-fast motors tasked with translocating DNA from which they must also remove DNA-bound protein roadblocks. Based on these results, we formulate a comprehensive mechanochemical model for SpoIIIE.
DOI: https://doi.org/10.7554/eLife.32354.001

## Introduction

Many cellular tasks are mechanical in nature, including nucleic acid/polypeptide translocation, nucleic acid strand separation, and chromosome segregation. Performing these tasks are a diverse range of molecular motor proteins, including the ring-shaped NTPases from the ASCE division of molecular motors (*Liu et al., 2015*, *Liu et al., 2014a*). These enzymes typically hydrolyze Adenosine Triphosphate (ATP) and utilize the free energy released upon hydrolysis to perform mechanical work (*Bustamante et al., 2004*).

Ring-shaped ATPases perform mechanical tasks by orchestrating the operation of their individual subunits (*Liu et al., 2014a*). Each ATPase subunit cycles through a series of chemical transitions (ATP binding, hydrolysis, ADP and Pi release) and mechanical events (track binding, power-stroke, motor resetting and release from the track). The coupling between chemical and mechanical transitions determines how individual subunits operate while the coordination between subunits determines how the entire ring ATPase functions. Understanding the operating principles of these molecular

machines requires a mechanistic model of ring ATPases at the level of individual subunits and the entire complex.

Here we used optical tweezers to interrogate the mechanism of DNA translocation by SpoIIIE, a homo-hexameric ring ATPase tasked with segregating the *B.subtilis* genome during sporulation (*Yen Shin et al., 2015*). Among ring ATPases, SpoIIIE and its *E.coli* homologue FtsK stand out as the fastest known nucleic acid translocases, pumping DNA at an astonishing 4000–7000 bp/s. (*Lee et al., 2012*; *Ptacin et al., 2006*). Previous studies of SpoIIIE/FtsK investigated how they bind DNA, what determines their translocation direction (*Lee et al., 2012*; *Levy et al., 2005*; *Ptacin et al., 2008*), how they displace or bypass DNA-bound protein roadblocks (*Crozat et al., 2010*; *Lee et al., 2014*), and how interaction with their track leads to DNA supercoiling during translocation (*Saleh et al., 2005*). In addition, we recently characterized SpoIIIE's inter-subunit coordination and presented evidence for a two-subunit translocation-escort model where one subunit actively translocates DNA while its neighbor passively escorts DNA (*Liu et al., 2015*). However, the detailed mechano-chemical coupling underlying the operation of ultra-fast ATPases like SpoIIIE/FtsK remains largely unknown.

## Results

### SpoIIIE generates up to 50 pN of mechanical force

Experiments were conducted on an instrument consisting of an optical trap and a micropipette as described previously (*Liu et al., 2015*). Briefly, SpoIIIE and DNA were immobilized separately on polystyrene beads (*Figure 1A*) and brought into proximity, allowing SpoIIIE to engage DNA. In the presence of ATP, SpoIIIE translocated DNA, shortening the tether between the two beads. Experiments were performed either in passive mode – where the trap position is fixed (*Figure 1B*), or in constant-force mode – where DNA tension is held constant (*Figure 1C*). At saturating [ATP] and low opposing force (5 pN), SpoIIIE translocated DNA at ~4 kbp/s (*Figure 1C*), in agreement with previous studies (*Liu et al., 2015*; *Ptacin et al., 2008*). Translocation rates measured in passive mode were in excellent agreement with those measured in constant-force mode (*Figure 1D*). We find that SpoIIIE can operate against forces up to 50 pN (*Figure 1B*), similar to other dsDNA translocases, including FtsK and the DNA packaging motors from bacteriophages T4, λ, and φ29 (*Fuller et al., 2007a*; *2007b*; *Saleh et al., 2004*; *Smith et al., 2001*).

### ATP mitigates force-induced slipping

To investigate SpoIIIE operation, we monitored translocation in passive mode. At sufficiently high opposing forces (20–40 pN), translocation trajectories are often interrupted by slips, presumably due to SpoIIIE losing grip of its DNA track (*Figure 1E*). Eventually, SpoIIIE can recover, re-engage the DNA, and resume translocation from a low force; consequently, the same motor can undergo many rounds of continuous translocation and slipping.

As is shown in *Figure 1—figure supplement 1A*, at saturating [ATP] (3 mM), SpoIIIE can undergo multiple rounds of pulling and slipping in passive mode, with a median slipping force of ~20 pN (*Figure 1—figure supplement 1B*). At low [ATP], the median slipping force drops below 15 pN (*Figure 1—figure supplement 1C*), suggesting that the nucleotide state modulates the strength of SpoIIIE-DNA interactions. To investigate how [ATP] affects slipping we conducted constant force experiments at 5–40 pN. At low [ATP] (0.25–0.50 mM) the slipping density increases sharply with opposing force, whereas at near-saturating [ATP] (1–3 mM) the slipping density is only weakly dependent on force (*Figure 1F*). Thus, binding of nucleotide to the motor appears to stabilize its grip on the DNA template. The slipping behavior of SpoIIIE as reported here is similar to that of the ATPase from the λ phage packaging motor (*delToro et al., 2016*) and it may also be a common feature of ASCE ring ATPases.

### The SpoIIIE power stroke is most likely driven by $P_i$ release

To probe how nucleotide binding is coordinated among motor subunits, we measured the pause-free SpoIIIE velocity at 3–50 pN of opposing force and 0.25–5.00 mM [ATP] (*Figure 2A*) in passive

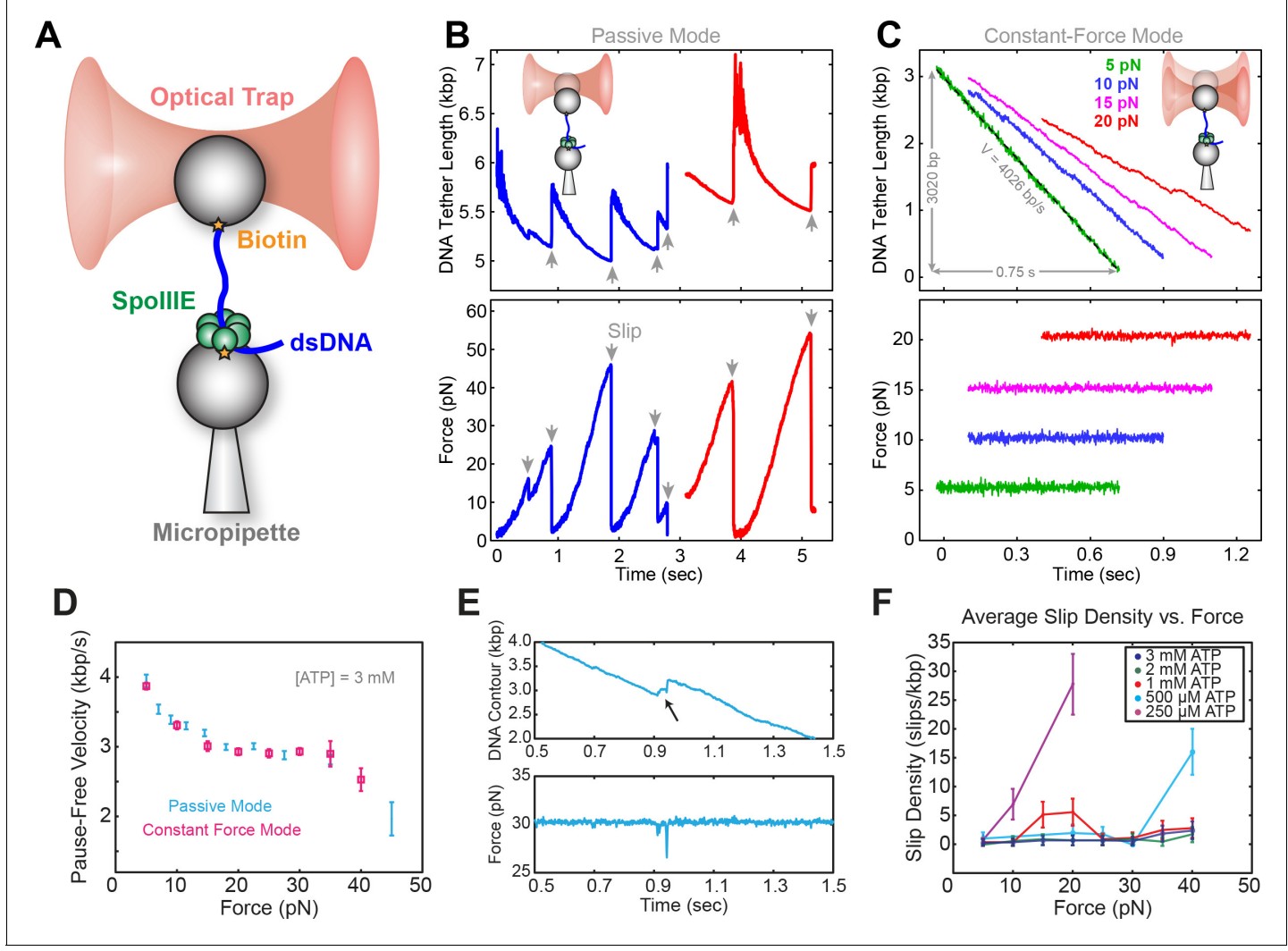

**Figure 1.** Optical tweezer experimental geometry in constant force and passive mode. (a) Optical tweezer geometry. (b) Representative single-molecule traces of SpoIIIE translocation in passive mode. The trap position is fixed and as SpoIIIE pulls the bead out of the trap, the force on the trapped bead increases. (c) Representative single-molecule traces of SpoIIIE translocation in constant force mode. The optical trap position is continuously adjusted to maintain a constant force on the trapped bead. (d) Comparison of pause-free velocity measured in constant force mode and passive mode at [ATP]=3 mM. Error-bars represent the standard error of the mean (SEM). (e) Trace displaying a slip in constant force mode. (f) Slip density at different opposing force and [ATP]. Error bars represent the square root of the number of events.
DOI: https://doi.org/10.7554/eLife.32354.002
The following figure supplement is available for figure 1:

**Figure supplement 1.** Slipping behavior of SpoIIIE.
DOI: https://doi.org/10.7554/eLife.32354.003

mode (*Figure 2—figure supplement 1A*). Fitting the pause-free velocity versus [ATP] to the Hill equation, yields a value for the Hill coefficient consistent with unity over a wide range of forces (5–30 pN) (*Figure 2—figure supplement 1B–C*). There are two means to achieve $n_{Hill} \approx 1$ for a multi-subunit ATPase: (i) subunits turnover ATP independently of each other in an uncoordinated fashion; or (ii) subunits turnover ATP sequentially, but consecutive binding events are separated by an irreversible transition so only one subunit can bind nucleotide at any time, resulting in an *apparent* lack of cooperativity (*Chemla et al., 2005*). We recently found that SpoIIIE pauses when two neighboring subunits each bind a non-hydrolyzable ATP analog (*Liu et al., 2015*). This result is inconsistent with scenario (i) outlined above because an uncoordinated mechanism should enable several subunits to bind ATP analogs while the remaining subunits continue translocating. We conclude that SpoIIIE

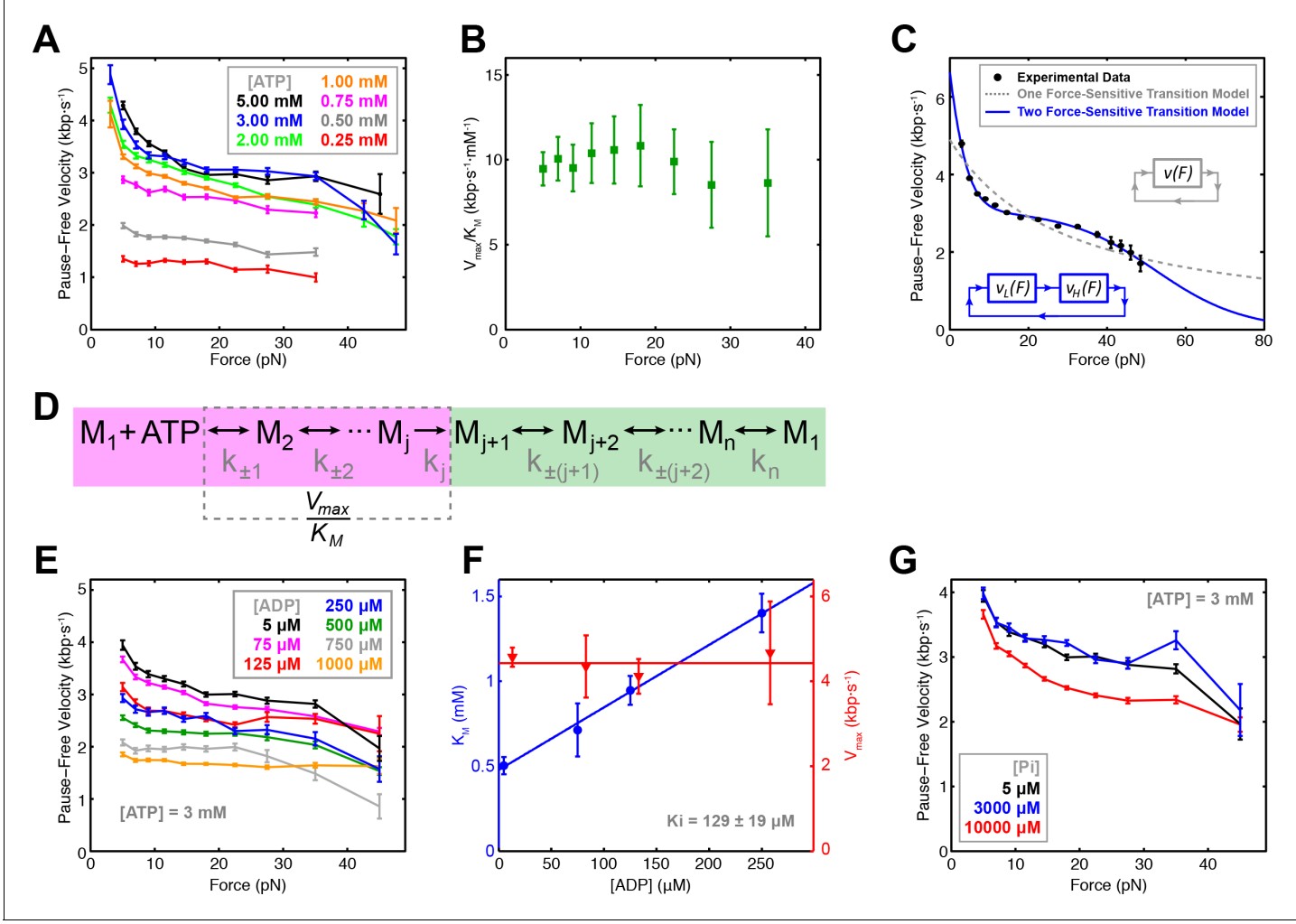

**Figure 2.** Force-velocity dependence displayed of SpoIIIE. (a) Pause-free translocation velocity versus opposing force at various [ATP], 5 μM ADP, and 5 μM $P_i$. Error-bars represent the SEM. (b) Hill coefficient derived from fitting translocation velocity versus [ATP] at various opposing forces. Error bars represent the standard error of the fit (SEF). (c) Pause-free velocity versus opposing force compiled from data at 5, 3, and 2 mM ATP. Error-bars represent the SEM. Gray and blue curves represent fits to the two different models depicted in the inset. Analytic expressions and fit parameters for the models are given in *Figure 2—figure supplement 2*. (d) Generalized kinetic cycle for an ATPase subunit. The first block consists of all rate constants $k_{\pm1}$, $k_{\pm2}$, . . . up to the first irreversible transition $k_j$ (purple). The second block comprises the remaining rate constants (green). (e) Pause-free velocity versus opposing force at various [ADP] and 3 mM ATP. (f) $V_{max}$ and $K_M$ values as a function of [ADP] at low opposing force (5 pN). Solid lines are fits to a competitive inhibition model, $K_i = 129 \pm 19$ μM. Error-bars represent the SEF. (g) Pause-free velocity versus opposing force under high [$P_i$] conditions and 3 mM ATP. Error bars represent the SEM.

DOI: https://doi.org/10.7554/eLife.32354.004

The following figure supplements are available for figure 2:

**Figure supplement 1.** Michaelis-Menten fits to SpoIIIE translocation.
DOI: https://doi.org/10.7554/eLife.32354.005
**Figure supplement 2.** Diagrams of mechanochemical models with two force-dependent transitions.
DOI: https://doi.org/10.7554/eLife.32354.006
**Figure supplement 3.** SpoIIIE slipping behavior in presence of ADP.
DOI: https://doi.org/10.7554/eLife.32354.007
**Figure supplement 4.** SpoIIIE pause detection and pause removal.
DOI: https://doi.org/10.7554/eLife.32354.008

subunits bind ATP sequentially one subunit at a time. This coordination scheme enforces the well-defined subunit firing order required for SpoIIIE to track the backbone of one DNA strand as we previously showed (*Liu et al., 2015*).

To determine which chemical transition is coupled to the power stroke we investigated how force affects $V_{max}$ and $K_M$ determined from Michaelis-Menten fits. Although both $V_{max}$ and $K_M$ decrease with force, $V_{max}/K_M$ is largely force-independent (*Figure 2B*). To understand this result, consider a generalized ATPase cycle consisting of two kinetic blocks separated by an irreversible transition $k_j$ (*Figure 2D*). We hypothesize that ATP tight binding (the transition that commits the ATPase to perform hydrolysis) is the irreversible transition that separates the kinetic blocks in *Figure 2D*, as has been proposed for other ring ATPases (*Chemla et al., 2005*; *Moffitt et al., 2009*; *Sen et al., 2013*). $V_{max}/K_M$ depends on ATP docking/undocking rates ($k_{\pm1}$) and the rates of all kinetic transitions reversibly connected to ATP docking ($k_{\pm2}$, $k_{\pm3}$...) up to the first irreversible transition $k_j$, (*Figure 2D*, purple) (*Keller and Bustamante, 2000*). The observed force-independence of $V_{max}/K_M$ indicates that ATP docking or any transition reversibly connected to it (*Figure 2D*, purple) cannot be the force-generating transition (*Keller and Bustamante, 2000*). Our observation that SpoIIIE is less force-sensitive at low [ATP], where nucleotide binding is rate-limiting, also suggests that ATP binding is not coupled to the power stroke. If ATP binding were coupled to the power stroke, at low [ATP] conditions the motor would be more, not less force sensitive. Therefore, the force-generating transition must occur in the second block of the generalized kinetic cycle (*Figure 2D*, green). It is unlikely that ATP hydrolysis drives the power stroke because the cleavage of the γ-phosphate upon hydrolysis does not release sufficient free energy (*Oster and Wang, 2000*). Therefore, ADP or $P_i$ release — both of which are located in the second kinetic block (*Figure 2D*, green) — must be responsible for force generation.

To distinguish between these possibilities, we quantified the inhibitory effect of ADP and Pi on translocation. We found that pause-free velocity decreased with increasing [ADP] (*Figure 2E*, *Figure 2—figure supplement 1D*). The apparent $K_M$ increases linearly with [ADP] whereas $V_{max}$ is independent of [ADP] (*Figure 2F*, *Figure 2—figure supplement 1E*), indicating that ADP is a competitive inhibitor to ATP binding with a dissociation constant $K_d = 129 \pm 19$ µM. In contrast, pause-free velocity is largely unaffected by increasing [$P_i$], decreasing by only ~12% at the highest $P_i$ concentration tested (10 mM) (*Figure 2G*), indicating that phosphate release is largely irreversible with a $K_d \gg 10$ mM. Given these $K_d$ values, we estimated the change in free energy upon Pi and ADP release $\Delta G_{P_i} > 7.6$ $k_BT$ and $\Delta G_{ADP} \sim 3.2$ $k_BT$ in a buffer containing 5 µM $P_i$ and 5 µM ADP (*Chemla et al., 2005*) (see Materials and methods). Given the estimated SpoIIIE step size of 2 bp (*Liu et al., 2015*), and a maximum generated force of ~50 pN, each SpoIIIE power-stroke requires at least 8.2 $k_BT$ of free energy (see Materials and methods). We conclude that phosphate release is the only chemical transition capable of driving the power stroke of SpoIIIE, similar to what has been proposed for the φ29 packaging motor (*Chemla et al., 2005*), and the ClpX ring ATPase (*Sen et al., 2013*).

## The SpoIIIE cycle contains at least two force-dependent kinetic rates

At near-saturating [ATP], SpoIIIE exhibits a bi-phasic force-velocity dependence: the pause-free velocity drops between 5 and 15 pN, remains relatively force-insensitive between 15 and 40 pN, then decreases again beyond 40 pN (*Figure 2A*). The large error-bars associated with velocity measurements at 40–50 pN are due to the limited amount of data that could be acquired at very high forces (*Table 1*). As a result it is challenging to assess the steepness of the velocity drop-off at high forces (*Figure 2A*, black, blue, and green curves). To overcome the limited data coverage at high forces and to better visualize the force-velocity behavior of SpoIIIE, we combined the data at near-saturating [ATP] (2, 3, 5 mM) into a consolidated curve (*Figure 2C*) that clearly displays the bi-phasic force-velocity dependence (see Materials and methods). Since the error-bars for the near-saturating [ATP] datasets partially overlap, especially in the high-force regime we reasoned that generating a consolidated force-velocity curve would not introduce significant bias.

A model with a single force-sensitive transition is inconsistent with the bi-phasic force-velocity dependence we observe for SpoIIIE because: (i) it predicts a monotonic decrease in velocity with force and poorly fits the data (*Figure 2C*, dashed gray curve), and (ii) it requires more free energy per power stroke than is released by hydrolyzing one ATP (see Materials and methods).

**Table 1.** Length of DNA (kbp) translocated at different forces and ATP concentrations. Related to *Figure 2C*.

| [ATP] (μM) | Force Interval (pN) | | | | | | | | | | | |
| --- | --- | --- | --- | --- | --- | --- | --- | --- | --- | --- | --- | --- |
| | 2–4 | 4–6 | 6–8 | 8–10 | 10–13 | 13–16 | 16–20 | 20–25 | 25–30 | 30–40 | 40–45 | 45–50 |
| 5000 | 105.0 | 98.2 | 75.1 | 61.5 | 77.6 | 60.3 | 51.4 | 34.8 | 16.2 | 11.8 | 1.0 | 0.2 |
| 3000 | 44.6 | 40.1 | 33.7 | 30.0 | 42.4 | 38.6 | 45.5 | 42.6 | 23.7 | 16.3 | 1.9 | 0.6 |
| 2000 | 39.0 | 38.9 | 34.6 | 32.1 | 45.4 | 41.5 | 47.9 | 41.3 | 21.0 | 12.1 | 1.8 | 0.9 |
| 1000 | 76.5 | 66.3 | 58.6 | 54.9 | 77.1 | 70.1 | 76.9 | 60.2 | 29.6 | 19.1 | 1.6 | 0.5 |
| 750 | 23.8 | 18.4 | 14.4 | 12.3 | 16.6 | 14.1 | 14.1 | 10.1 | 5.2 | 3.6 | 0.3 | 0 |
| 500 | 44.1 | 43.7 | 37.8 | 33.3 | 42.4 | 34.9 | 31.5 | 19.6 | 8.1 | 3.7 | 0 | 0 |
| 250 | 22.4 | 20.4 | 18.4 | 17.4 | 23.4 | 17.1 | 13.5 | 9.9 | 4.2 | 2.0 | 0 | 0 |

DOI: https://doi.org/10.7554/eLife.32354.009

At least two force-sensitive transitions are needed to rationalize SpoIIIE's force-velocity dependence: the first should capture the motor's sensitivity to force at low loads (<15 pN), the second should describe the motor's sensitivity to force at high loads (>40 pN). By introducing a second force-dependent transition in the mechanochemical cycle (*Figure 2—figure supplement 2A*), our model accurately captures the bi-phasic force-velocity dependence exhibited by SpoIIIE (*Figure 2C*).

## Pausing is in kinetic competition with translocation and ATP binding

At low [ATP], SpoIIIE exhibits spontaneous pausing (*Figure 3A*). We find that pause density increases dramatically as pause-free velocity drops (*Figure 3B*), suggesting that pausing and translocation are in kinetic competition. This observation is consistent with a model where the pause state is off the main translocation pathway, similar to what has been observed for the λ phage packaging motor (*delToro et al., 2016*) and the ClpX protein unfoldase (*Maillard et al., 2011*). To determine where in the mechanochemical cycle the off-pathway pause state is located, we analyzed SpoIIIE's pausing at various [ATP] and forces. We find that pause density increases drastically at low [ATP] (*Figure 3C*) indicating that pausing is in kinetic competition with nucleotide binding. In other words, SpoIIIE enters a pause when a subunit is awaiting ATP binding. We also found that the mean pause duration is inversely proportional to [ATP] (*Figure 3D*) suggesting that SpoIIIE exits the paused state by binding nucleotide. Pause durations at a given [ATP] are exponentially distributed (*Figure 3D*, inset) indicating that pause duration is governed by a single-rate limiting event—presumably the motor binding an ATP molecule. Due to limitations governed by our time resolution and experimental noise, we could not accurately detect pauses shorter than ~50 msec (Materials and methods); we therefore estimated the mean pause duration by fitting the duration of the observed pausing events to a single-exponential. Finally, the fact that the pause density and the estimated pause duration does not depend on force at low [ATP] (where ATP binding is rate-limiting) (*Figure 3E–F*) suggests that the pause state is not reversibly connected to the force-generating transition ($P_i$ release).

## Discussion

### The mechanochemical cycle of an individual SpoIIIE subunit

Based on the results above, we propose a minimal mechanochemical model for a single SpoIIIE subunit (*Figure 4A*). The ADP-bound state (gray) is reversibly connected to the Apo state (white), which is reversibly connected to the ATP-loosely-docked state (light green). Here ADP acts as a competitive inhibitor to ATP binding, as observed experimentally. The ATP-loosely-docked state is irreversibly connected to the ATP-tightly-bound state (dark green), ensuring that $V_{max}/K_M$ is force-insensitive. Hydrolysis is depicted as a reversible process between the ATP-tightly-bound state and the transition state ADP·$P_i$ (blue). Finally, $P_i$ release is depicted as an irreversible process that drives the 2 bp power-stroke.

When does SpoIIIE make and break its DNA contacts? We previously found that the strength of the motor-DNA interaction is highest in the ATPγS-bound state, moderate in the ADP-bound state,

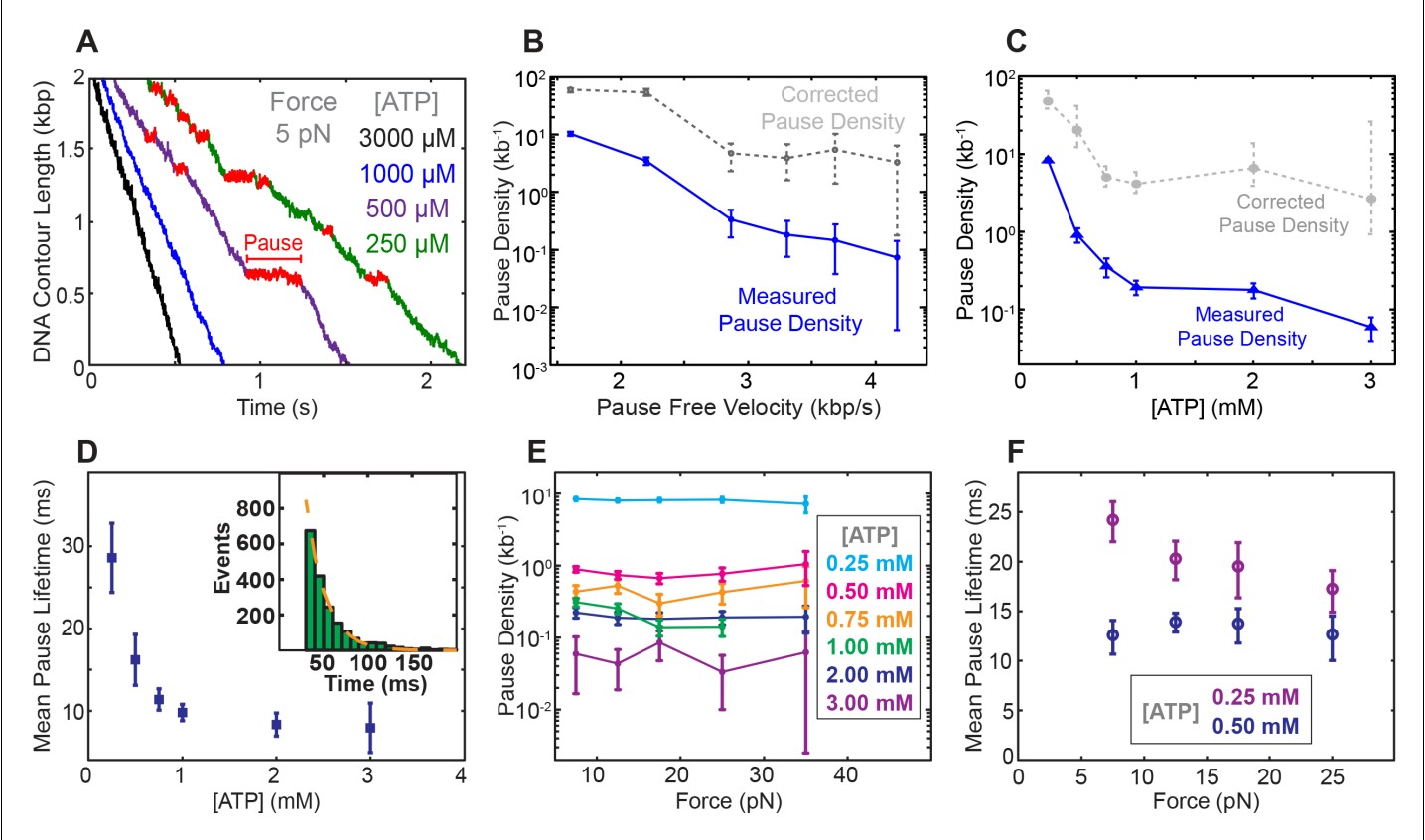

**Figure 3.** Characterization of spontaneous pausing by SpoIIIE. (**a**) Examples of SpoIIIE translocation trajectories acquired at low force (5 pN) and various ATP concentrations with detected pauses highlighted in red. (**b**) Measured pause density (solid lines) and corrected pause density (dashed lines) accounting for the missed pauses versus pause-free velocity at 5 pN. (**c**) Measured and corrected pause densities versus ATP concentration at 5 pN. (**d**) The mean pause lifetime calculated by fitting the distribution of pause durations to a single exponential (see inset). Error-bars represent the SEF. (Inset) Distribution of pause durations at 250 μM [ATP] (green) fit to a single-exponential decay (dashed line). The mean pause lifetime estimates at high [ATP] are less accurate due to the low number of detectable pauses. (**e**) Measured pause density versus opposing force at various [ATP]. (**f**) Mean pause lifetimes versus opposing force at the two lowest [ATP], where the number of pauses was sufficiently high to accurately estimate the lifetimes from fits. Error bars from fits represent 95% CI from fits. Error bars of pause density estimated from square root of the number of pause events.

DOI: https://doi.org/10.7554/eLife.32354.010

and lowest in the Apo state (*Liu et al., 2015*). Given that nucleotides strengthen the motor-DNA interactions, we propose that each SpoIIIE subunit has to bind ATP first before it engages the DNA and the motor-DNA interaction is established during ATP docking or during tight–binding (*Figure 4A*, green box). Since the ADP-bound and the Apo states have the weakest affinity for DNA we propose that each subunit breaks its contacts with DNA after reaching the ADP-bound state or the Apo state (*Figure 4A*, yellow box).

## Force-induced slipping: implications for the two-subunit translocation-escort mechanism

We previously provided evidence for a model where two subunits contact the DNA at adjacent pairs of phosphates on the same strand: while one subunit executes the power-stroke and translocates 2 bp, the other escorts the DNA (*Liu et al., 2015*). This mechanism enables the motor to operate processively with non-consecutive inactive subunits, and the escorting subunit may function as a backup should the translocating subunit lose its grip on DNA during the power-stroke. In the present study, we find that slipping probability can be increased by either large opposing force or low ATP conditions or a combination of both.

Based on the insights from the slipping data, we propose a revised model of the one proposed before (*Liu et al., 2015*) (*Figure 4B*). After executing the power-stroke, the translocating subunit (A)

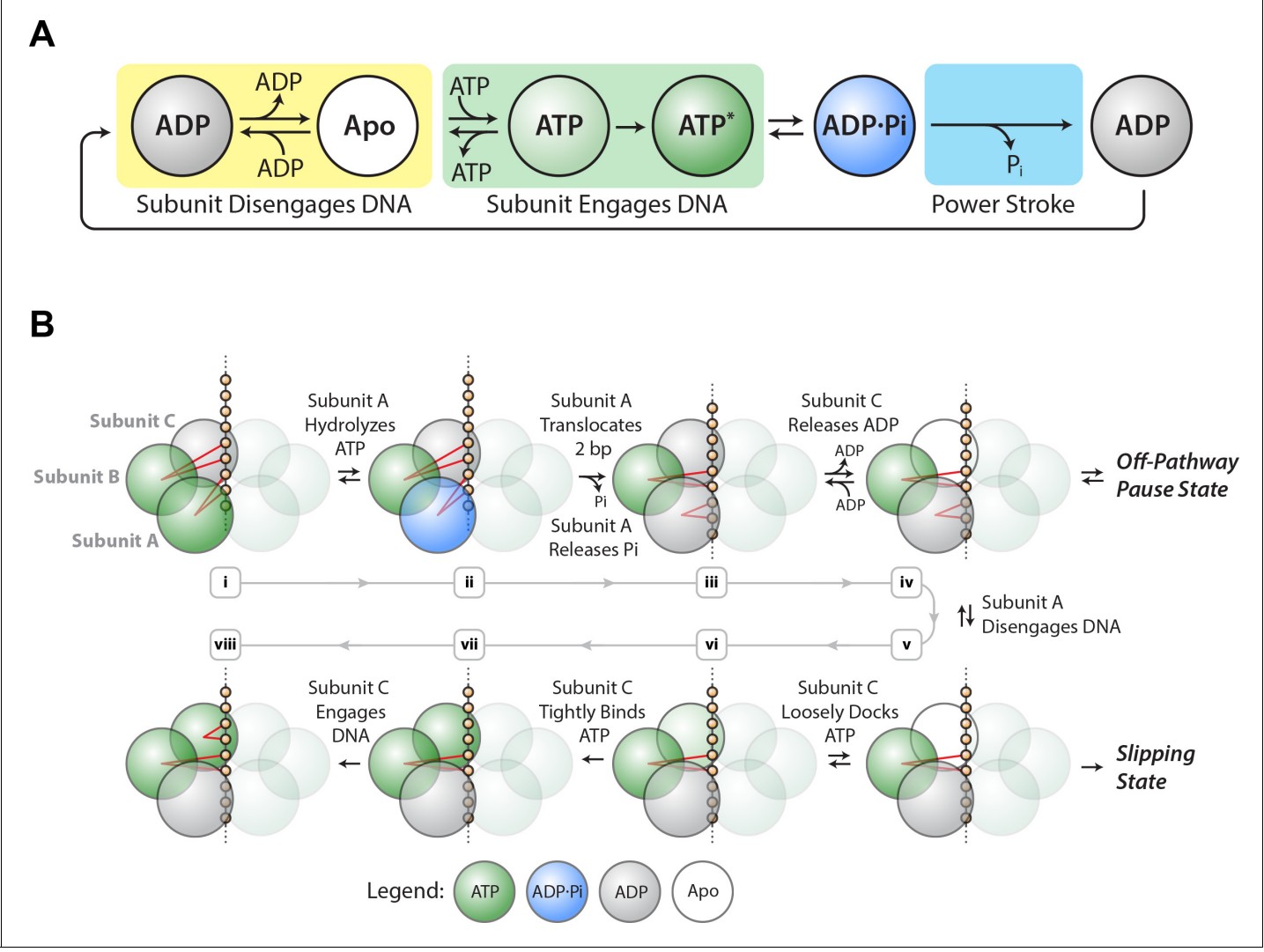

**Figure 4.** SpoIIIE mechanochemistry model. (a) Mechanochemical cycle for a single SpoIIIE subunit. (b) Mechanochemical cycle for the entire SpoIIIE homo-hexamer.

DOI: https://doi.org/10.7554/eLife.32354.011

disengages DNA, the escorting subunit (B) maintains its grip on DNA while the next subunit (C) first binds ATP and then engages the DNA (*Figure 4Biii–vi*). After this hand-over, subunits B and C become the new translocating and escorting subunits respectively and this cycle continues around the ring. The motor is most vulnerable to slipping while B is the only subunit anchoring the hexamer to the DNA backbone (*Figure 4Bv*). At high [ADP] and low [ATP], subunit C spends more time in the ADP-bound or Apo state, lengthening the time in which subunit B is the only one anchoring the motor onto DNA, and increasing the slipping probability (*Figure 2—figure supplement 3*). When the escorting and translocating subunits are both contacting DNA, the likelihood of force-induced slipping is significantly diminished.

In our model ADP release happens before ATP binding (see subunit C in *Figure 4Biii–iv*). Since ADP acts as a competitive inhibitor to ATP binding, in our model ADP release and ATP docking in the same subunit are connected via reversible transitions as depicted in *Figure 4Biii–vi*. It is unclear what triggers ADP release, however studies of related ATPases show that ADP release is highly coordinated among subunits, triggered for example by the binding of ATP in the adjacent subunit (*Chistol et al., 2012*).

## Off-pathway pausing: timing and implications

We found that pause density is inversely proportional to pause-free velocity (*Figure 3B*), indicating that pausing is an off-pathway process in kinetic competition with translocation. The observation that pausing is more likely at low [ATP] (*Figure 3E*) suggests that SpoIIIE pauses when a subunit is awaiting the binding of ATP. At the same time, we do not observe frequent slipping from paused states. We speculate that SpoIIIE enters off-pathway pauses from the state depicted in *Figure 4Biv* – after subunit A translocated but before DNA is handed to subunit C. At this stage, subunit C is poised to bind ATP. At low [ATP], if subunit C takes a long time to bind nucleotide, SpoIIIE may transition into an off-pathway pause state while gripping the DNA with two subunits. Such an allosteric sensing mechanism would prevent the motor from prematurely initiating the slip-prone DNA handover (*Figure 4Bv*). This speculative regulatory mechanism for SpoIIIE is reminiscent of the allosteric regulation of the φ29 viral packaging motor, which senses when the capsid is nearly full and enters into long-lived pauses allowing DNA inside the capsid to relax before packaging can restart (*Berndsen et al., 2015*; *Liu et al., 2014b*). During chromosome segregation in *B. subtilis*, the local [ATP] near active SpoIIIE complexes could fluctuate on short time-scales. A drop in local [ATP] could force the motor to pause thus preventing the slip-prone handover until [ATP] rises to levels optimal for SpoIIIE operation.

## Bi-phasic velocity-vs-force dependence and its implications

In addition to SpoIIIE, a bi-phasic force-velocity dependence has been reported for several other DNA translocases, including FtsK (*Saleh et al., 2004*), and the λ and T4 phage packaging motors (*Fuller et al., 2007b*; *Migliori et al., 2014*). To explain this unusual behavior we propose a mechano-chemical cycle containing two sequential force sensitive transitions (*Figure 2C*, blue inset): one that is highly sensitive to force and saturates at >15 pN, causing the velocity decrease up to ~15 pN; and another that is less sensitive to force, leading to the velocity drop beyond ~40 pN.

The force-velocity dependence observed at high forces reflects the fact that the force-generating transition becomes rate-limiting at sufficiently high mechanical loads. We speculate that the force-velocity dependence observed at low force reflects a load-induced motor deformation that slows down a kinetic transition distinct from the power stroke, and this deformation saturates at ~15 pN. The force-independence of $V_{max}/K_M$ suggests that this transition occurs after ATP tight binding (ATP hydrolysis, ADP release, or another transition distinct from $P_i$ release). Fitting the force-velocity data to the model predicts a velocity of ~6.5 kbp/s under no load (*Figure 2—figure supplement 2A*), in agreement with the zero-force maximum velocity of FtsK – SpoIIIE's homologue in *E. coli* (*Lee et al., 2012*).

Alternatively the biphasic force-velocity dependence can be rationalized by two force-generating transitions in a branched model (*Figure 2—figure supplement 2B*), where the motor executes two alternative power strokes with different force-sensitivities (see Materials and methods). The branched model describes a motor that can perform one power stroke at the exclusion of the other, alternating between two distinct mechanochemical cycles, a property that has not been demonstrated for any known molecular motor. Thus we disfavor this model for SpoIIIE.

Interestingly, the in vivo SpoIIIE rate is markedly slower (~1–2 kb/sec) than its in vitro rate (~4–5 kb/sec) (*Burton et al., 2007*; *Ptacin et al., 2008*). This discrepancy can be explained by the fact that DNA-bound proteins act as physical barriers to SpoIIIE in vivo effectively creating an opposing force (*Marquis et al., 2008*). Although we do not have an accurate estimate of the opposing forces experienced by individual motors in vivo, we expect that they are up to tens of pN because such forces are needed to disrupt protein-DNA interactions in vitro (*Dame et al., 2006*). DNA-bound protein barriers are obstacles encountered by most DNA translocases - viral packaging motors, helicases, chromosome segregases - all of which travel on tracks with multiple protein roadblocks that hinder translocation. Our study provides insight into how ultra-fast ring ATPases like SpoIIIE and FtsK may respond to a variety of physical and chemical challenges inside the cell, such as decreasing translocation velocity when encountering opposing forces and roadblocks, slipping at high forces, and pausing at low ATP concentrations.

# Materials and methods

## Sample preparation

Recombinant SpoIIIE, dsDNA substrates, and polystyrene beads were prepared as described before (*Liu et al., 2015*).

## Data analysis

Tether tension and extension were converted to contour length using the Worm-Like-Chain approximation (*Baumann et al., 1997*).

Pauses were detected using a modified Schwartz Information Criterion (mSIC) method (*Maillard et al., 2011*) (see *Figure 2—figure supplement 4* panel C). The number and duration of pauses missed by this algorithm were inferred by fitting the pause duration distribution to a single exponential with a maximum likelihood estimator. For an in-depth explanation of pause analysis and missed pause estimation please refer to the Data Analysis and Methods section of our previous study (*Liu et al., 2015*). For 0.25 mM and 0.50 mM ATP, pauses of 50 ms or longer were reliably detected and removed, whereas at higher [ATP] pauses of 30 ms or longer were removed. The removal of detected pauses had only a minor effect on the measured translocation velocity – compare panels A and B of *Figure 2—figure supplement 4*.

After removing the detected pauses, the translocation velocity was computed by fitting the data to a straight line. For passive-mode data, single-molecule trajectories were partitioned into segments spanning 2 pN, and the velocity was computed for each segment. The data for force-velocity measurements was collected in passive mode where the opposing force increases gradually as the motor translocates DNA. The individual translocation traces were segmented into windows spanning 2 pN each and the translocation velocity was computed for each force window. To generate the consolidated force-velocity curve at near-saturating [ATP] (*Figure 2C*) we pooled the velocity measurements from individual 2-pN force windows and then binned them.

## Estimating free energy of product release

To estimate the free energy of product release, consider the simplified kinetic scheme $E \cdot P \rightleftharpoons E + P$ where the enzyme (E) can release or bind its product (P) with a forward and reverse rate $k_{rel}$ and $k_{bind}$ respectively. We can define the rate of phosphate release as $k_{rel} = k_{-p}$ and the rate of phosphate binding as $k_{bind} = k_p \cdot [P_i]$ where $k_{-p}$ and $k_p$ are the first and second-order rate constants for phosphate release and binding respectively. The free energy change corresponding to phosphate release is given by $\Delta G_{Pi} = -k_BT \cdot \ln(k_{rel}/k_{bind}) = -k_BT \cdot \ln(k_{-p}/k_p \cdot [P_i])$ (*Chemla et al., 2005*). Since concentrations of phosphate as high as 10 mM do not significantly affect SpoIIIE's translocation velocity, then $k_{rel}$ must be significantly higher than $k_{bind}$ at $P_i$ concentrations of 10 mM or less (i.e., $k_{-p} >> k_p \cdot [10 \text{ mM}] >> 1$). From these inequalities we can infer that $k_{-p} /(k_p \cdot [5 \text{ μM}]) > 2000$ and therefore we can set a lower bound for the free energy of $P_i$ release as $\Delta G_{Pi} > -7.6 \ k_BT$ in a buffer containing 5 μM $P_i$.

Similarly, we used the equilibrium dissociation constant for ADP ($K_{ADP} = 129 \pm 19$ μM) to estimate the change in free energy associated with ADP release: $\Delta G_{ADP} \sim 3.2 \ k_BT$ in standard buffer conditions ([ADP]=5 μM). Given the estimated SpoIIIE step size of 2 bp (*Graham et al., 2010*; *Liu et al., 2015*; *Massey et al., 2006*) and the fact that SpoIIIE can translocate DNA against forces as high as 50 pN each power-stroke requires a change in free energy of at least 50 pN $\cdot$ 2 bp $\cdot$ 0.34 nm/bp = 34 pN$\cdot$nm = 8.2 $k_BT$.

## Mechanochemical model with a single force-generating transition

Note that this model as well as the linear/branched models described in the next section assume Arrhenius-like force-dependent terms. We cannot rule out non-Arrhenius type force-dependences which could also lead to a velocity reduction at higher forces – for example force-induced decoupling of ATP turnover from DNA translocation. In a hypothetical case force applied to the DNA could deform the ATPase such that the motor loses its grip on DNA in a force-dependent manner, leading to non-productive power strokes and lower net translocation velocity.

A mechanochemical model with a single force-generating transition predicts that at saturating [ATP], the motor velocity (*V*) depends on the external load (*F*) as $V(F) = \frac{V_{max}}{(1-p)+p \cdot \exp\left(\frac{F\Delta x^{\ddagger}}{k_BT}\right)}$ (*Wang et al.,*

*1998*). Here *Vmax* is the maximum velocity at zero force, $\exp\left(\frac{F\Delta x^{\ddagger}}{k_B T}\right)$ is an Arrhenius-like term describing how the external load slows down the force-generating transition, *p* is the fraction of the total mechanochemical cycle time that the motor spends in the force-generating transition at zero force, (1-*p*) captures all the force-independent transitions from the motor's cycle, $k_B T$ is the Boltzmann constant times the temperature, and $\Delta x^{\ddagger}$ is the distance to the transition state for the force-generating transition.

Fitting the consolidated force-velocity curve to the model above produces a very poor fit to the data (*Figure 2C*, dashed gray curve) (*Vmax* = 4.2 ± 0.4 kbp/s), $\Delta x^{\ddagger}$ = 0.07 ± 0.02 nm, p ≈ 1), and most importantly predicts a monotonic decrease in velocity with force that does not capture the bi-phasic force-velocity dependence exhibited by SpoIIIE. Furthermore, extrapolating the fit to higher forces predicts large translocation velocities (>300 bp/sec) for loads over 400 pN. Considering that the likely step size of SpoIIIE is 2 bp per nucleotide hydrolyzed (*Liu et al., 2015*), a stall force above 400 pN requires that the motor generate at least 400 pN · 2 bp · 0.34 nm/bp = 270 pN·nm of work per power-stroke—more than two and a half times the ~110 pN•nm of free energy available from ATP hydrolysis in our experiments.

## Deriving expressions for the branched and linear models of force-velocity dependence

We considered two broad classes of kinetic models that can capture the bi-phasic velocity dependence on force: branched models and linear models. In each case the average time needed to complete one cycle can be computed given the rate of ATP binding (which is proportional to [ATP] with the proportionality constant $\alpha$), the rates of the two force-sensitive kinetic transitions ($k_A$ and $k_B$), and the net compound rate of all remaining kinetic transitions that are force-insensitive ($k_0$). The average cycle completion time for the branched model shown in *Figure 2—figure supplement 2B*, $\tau_{branched}$, can be written as follows:

$$\tau_{branched} = p_A \cdot \frac{1}{k_A} + p_B \cdot \frac{1}{k_B} + \frac{1}{\alpha \cdot [ATP]} + \frac{1}{k_0} \tag{1}$$

Here $p_A$ and $p_B$ are the probabilities that the cycle proceeds through each of the two force-sensitive branches (*Figure 2—figure supplement 2B*). For simplicity we assumed an Arrhenius-like dependence on force *F* for $p_A$.

$$p_A = p_0 \cdot e^{-\frac{F \cdot \Delta x_C^{\dagger}}{k_B T}}$$

$$p_B = 1 - p_A = 1 - p_0 \cdot e^{-\frac{F \cdot \Delta x_C^{\dagger}}{k_B T}} \tag{2}$$

The rates for the two force-sensitive transitions are given by $k_A$ and $k_B$ respectively, each with an Arrhenius-like dependence on force *F* as shown below. Here $k_B T$ is the product of the Boltzmann constant and the temperature, $k_{A0}$ and $k_{B0}$ are the rates at zero force, and $\Delta x^{\dagger}_A$ and $\Delta x^{\dagger}_B$ are the distances to the transition state for the two force-sensitive branches.

$$k_A(F) = k_{A0} \cdot e^{-\frac{F \cdot \Delta x_A^{\dagger}}{k_B T}}$$

$$k_B(F) = k_{B0} \cdot e^{-\frac{F \cdot \Delta x_B^{\dagger}}{k_B T}} \tag{3}$$

Each of the two force-sensitive transitions represents a power-stroke with step-sizes $d_A$ and $d_B$ respectively. Therefore, the average step size for the branched cycle $d_{branched}$ is given by:

$$d_{branched} = p_A d_A + p_B d_B \tag{4}$$

We fit the force-velocity data in *Figure 2C* to the simplest branched model where $d_A = d_B = d_{branched} = d$. Note that the model in which $d_A \neq d_B$ also fits the data, but is less well-constrained. The average translocation velocity for the branched model is given by the following expression:

$$V_{branched} = \frac{d_{branched}}{\tau_{branched}} = \frac{d}{\frac{1}{k_0} + \frac{1}{\alpha \cdot [ATP]} + \frac{1}{k_{B0}} \cdot e^{\frac{F \cdot \Delta x_B^\dagger}{k_B T}} + p_0 \cdot e^{-\frac{F \cdot \Delta x_C^\dagger}{k_B T}} \cdot \left( \frac{1}{k_{A0}} \cdot e^{\frac{F \cdot \Delta x_A^\dagger}{k_B T}} - \frac{1}{k_{B0}} \cdot e^{\frac{F \cdot \Delta x_B^\dagger}{k_B T}} \right)} \tag{5}$$

In a similar fashion, an expression for the translocation velocity can be derived for the linear model depicted in **Figure 2C**.

$$\tau_{linear} = \frac{1}{k_L} + \frac{1}{k_H} + \frac{1}{\alpha \cdot [ATP]} + \frac{1}{k_0} \tag{6}$$

$$d_{linear} = d \tag{7}$$

$$V_{linear} = \frac{d_{linear}}{\tau_{linear}} \tag{8}$$

Here $k_L$ and $k_H$ are the rates of the force-sensitive transitions responsible for the drop in velocity at low forces (0–15 pN) and high forces (40 pN and above). $k_H$ represents the rate of the force-generating transition, i.e. phosphate release (most likely), and is given by a simple Arrhenius-like dependence:

$$k_H(F) = k_{H0} \cdot e^{-\frac{F \cdot \Delta x_H^\dagger}{k_B T}} \tag{9}$$

As described in the main text, $k_L$ saturates at a certain force (~15 pN), and could be written as follows:

$$k_L(F) = k_{L0} \cdot \left( 1 + \beta \cdot e^{-\frac{F \cdot \Delta x_L^\dagger}{k_B T}} \right) \tag{10}$$

The final expression for $V_{linear}$ is:

$$V_{linear} = \frac{d}{\frac{1}{k_0} + \frac{1}{\alpha \cdot [ATP]} + \frac{1}{k_{L0} \cdot \left( 1 + \beta \cdot \exp\left( -F \cdot \Delta x_L^\dagger / (k_B T) \right) \right)} + \frac{1}{k_{H0}} \cdot \exp\left( F \cdot \Delta x_H^\dagger / (k_B T) \right)} \tag{11}$$

## Fitting the consolidated force-velocity curve to the linear model

The linear model provides two values for the distance to the transition state, $\Delta x_H^\dagger = 0.4 \pm 0.2$ $nm$ at high forces and $\Delta x_L^\dagger = 1.3 \pm 0.5$ $nm$ at low forces. A typical energy landscape for a molecular motor contains both a chemical axis, which captures the sequential chemical transitions a motor undergoes as it generates mechanical work, and a mechanical axis, which captures the physical movement of the motor along a distance coordinate (**Bustamante et al., 2004**). The distance to the transition state $\Delta x^\dagger$ is the distance the motor must move along the mechanical coordinate during the force-sensitive step in order to commit itself to stepping. If a motor directly couples a chemical transition to the force-generating step, in what is classically referred to as a 'power stroke', the motor will move approximately along a diagonal across the chemical and mechanical axes, and $\Delta x^\dagger$ would typically be $< \Delta x_{step}$, where $\Delta x_{step}$ is the distance the motor moves per step size. The value for $\Delta x_H^\dagger = 0.4 \pm 0.2$ $nm$ is smaller than and consistent with a 2 bp step size (0.68 nm) power stroke mechanism previously determined for SpoIIIE (**Liu et al., 2015**) and likely coupled to $P_i$ release.

The physical interpretation of the other distance to the transition state, $\Delta x_L^\dagger = 1.3 \pm 0.5$ $nm$ is less clear. We speculate that the initial decrease in velocity induced by force was attributed to motor deformation; the measured distance to the transition state $\Delta x_L^\dagger = 1.3 \pm 0.5$ $nm$ is consistent with transition state values observed for single-molecule unfolding of native state proteins, typically $\Delta x^\dagger < 2 nm$ for native state protein unfolding (**Bustamante et al., 2004**; **Elms et al., 2012**). However, the motor is clearly still active at forces >15 pN. It is possible that the measured value of $\Delta x_L^\dagger$ corresponds to the mechanical coordinates of a partial unfolding pathway.

## Acknowledgements

We thank SB Smith for instrument training and assistance; LM Alexander for experimental assistance, JY Shin, M Righini, B Onoa, and C Diaz for discussions. This work was supported by NIH grants R01GM032543 and the US. Department of Energy Office of Basic Energy Sciences Nanomachine Program under contract no. DE-AC02-05CH11231.

## Additional information

### Funding

| Funder | Grant reference number | Author |
|---|---|---|
| Howard Hughes Medical Institute | | Carlos Bustamante |
| National Institutes of Health | R01GM071552 | Carlos Bustamante |
| National Institutes of Health | R01GM032543 | Carlos Bustamante |
| U.S. Department of Energy | DE-AC02-05CH11231 | Carlos Bustamante |

The funders had no role in study design, data collection and interpretation, or the decision to submit the work for publication.

### Author contributions

Ninning Liu, Conceptualization, Data curation, Software, Formal analysis, Investigation, Visualization, Methodology, Writing—original draft, Writing—review and editing, Conducted the majority of single-molecule experiments, Prepared samples and analyzed the data, Wrote MATLAB code for data analysis; Gheorghe Chistol, Conceptualization, Data curation, Software, Formal analysis, Investigation, Visualization, Methodology, Writing—original draft, Writing—review and editing, Prepared samples and analyzed the data, Wrote MATLAB code for data analysis; Yuanbo Cui, Investigation, Writing—original draft, Collected data in constant-force mode; Carlos Bustamante, Conceptualization, Supervision, Funding acquisition, Writing—original draft, Project administration, Writing—review and editing

### Author ORCIDs

Ninning Liu http://orcid.org/0000-0002-8398-9584
Carlos Bustamante http://orcid.org/0000-0002-2970-0073

### Decision letter and Author response

Decision letter https://doi.org/10.7554/eLife.32354.014
Author response https://doi.org/10.7554/eLife.32354.015

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
