## [Decision Letter]

Thank you for submitting your article "Mechanochemical Coupling and Bi-Phasic Force-Velocity Dependence in the Ultra-Fast Ring ATPase SpoIIIE" for consideration by *eLife*. Your article has been reviewed by three peer reviewers, and the evaluation has been overseen by a Reviewing Editor and Anna Akhmanova as the Senior Editor. The following individual involved in review of your submission has agreed to reveal his identity: Douglas Smith (Reviewer #1).

The reviewers have discussed the reviews with one another and the Reviewing Editor has drafted this decision to help you prepare a revised submission.

Summary:

This is a very nice and comprehensive study of the mechanochemical kinetics of the SpoIIIE motor which should be of wide interest to those studying chromosome segregation motors and other related/homologous motors which may utilize some of the same mechanisms. The data are of high quality and measurements obtained spanning a wide range of relevant control parameters, which is a major feat given the difficulty of conducting these kind of single-molecule measurements. Evidence that many of the features, such as engaging DNA upon ATP binding and translocation upon phosphate release, are similar to those of the viral packaging motor is very interesting, as it suggests that widely applicable general principles have been discovered. However, the paper also reports finding of several varying features, such as the bi-phasic dependence of motor velocity on force and need to account in the kinetic model for a role of a second "escort" subunit, for example in accounting for the slipping behavior. The authors find that the motor can use two translocation modes: a 'low gear' that is slow but can withstand high force, and a 'high gear' that is susceptible to force but is extremely fast. The authors then speculate that SpoIIIE, and similar fast dsDNA translocases, use these two gears to remove proteins bound to the DNA during translocation.

Essential revisions:

1) The paper is a difficult read at times. Most of that difficulty is in support of very minor conclusions (e.g. all the material about branched vs linear enzymatic pathways and differentiating by fitting the biphasic f/vel curve, is quite involved, but doesn't really contribute to any major conclusions). Most of the major conclusions are based on quite simple (and therefore robust and believable) observations ("Velocity goes down very quickly with [Pi], but not with [ADP]"). Can the authors streamline the presentation so that the readers do not get lost while reading rather sophisticated arguments concerning minor points? In addition, the Introduction is incredibly short and rather vague. The Introduction should be rewritten to set up questions better and to prepare the readers so that they can appreciate the finding better.

2) A good bit of the paper discusses more quantitative modelling of the experimental data, which in turn relies on proper handling of the data. The conclusions of this part of the paper are not mentioned in the Abstract, presumably because the conclusions are a bit less robust. Removing pauses from motor translocation curves is known to be a difficult task, due to the presence of experimental noise along with limitations in time resolution leading to the likelihood of existence of pauses that are not measurable. It would be nice to see some traces with pauses labelled, to get a sense of what their algorithm counts as a pause. It would also be helpful to get a sense of the magnitude of some details of the pause analysis: First, they extrapolate the occurrence of short-time pauses from an exponential distribution; how many such "unseen" pauses are there relative to the visible ones? This bears on the various figures quantifying pause occurrences in Figure 3 (which count both seen and unseen pauses). Second, for Figure 2/2C, what does net velocity with pauses vs. force look like? It will be good to give the readers a sense of the sensitivity to pause-removal of the force/velocity dependence.

3) The main-text statement (subsection “The SpoIIIE Cycle Contains at Least Two Force-Dependent Kinetic Rates”) that the velocity "rapidly decreases again beyond 40 pN" in saturating ATP conditions seems to be contradicted by the data in Figure 2, where the sharp decrease seems only to be clear in the 3 mM data. The other ATP concentrations show instead a broad, continuous decrease in velocity rather than a clear high-force knee. Further, there is clearly large error in the high-force velocity points, as evidenced in Figure 2. This reduces confidence in the analysis based on the combination of the 2, 3 and 5 mM ATP data in Figure 2. On that note-- how are those data points even combined, given the different force values used for each [ATP] data set (as seen in Figure 2)?

---

## [Author Response]

Essential revisions:1) The paper is a difficult read at times. Most of that difficulty is in support of very minor conclusions (e.g. all the material about branched vs linear enzymatic pathways and differentiating by fitting the biphasic f/vel curve, is quite involved, but doesn't really contribute to any major conclusions). Most of the major conclusions are based on quite simple (and therefore robust and believable) observations ("Velocity goes down very quickly with [Pi], but not with [ADP]"). Can the authors streamline the presentation so that the readers do not get lost while reading rather sophisticated arguments concerning minor points? In addition, the Introduction is incredibly short and rather vague. The Introduction should be rewritten to set up questions better and to prepare the readers so that they can appreciate the finding better.

We agree that the “branched vs. linear model” argument is unnecessarily detailed. We greatly simplified this section of the paper and also edited Figure 2 to only illustrate only the preferred model. The alternative model is mentioned only briefly in the Discussion section. We have also rewritten the Introduction to better contextualize the findings of our study.

2) A good bit of the paper discusses more quantitative modelling of the experimental data, which in turn relies on proper handling of the data. The conclusions of this part of the paper are not mentioned in the Abstract, presumably because the conclusions are a bit less robust. Removing pauses from motor translocation curves is known to be a difficult task, due to the presence of experimental noise along with limitations in time resolution leading to the likelihood of existence of pauses that are not measurable. It would be nice to see some traces with pauses labelled, to get a sense of what their algorithm counts as a pause. It would also be helpful to get a sense of the magnitude of some details of the pause analysis: First, they extrapolate the occurrence of short-time pauses from an exponential distribution; how many such "unseen" pauses are there relative to the visible ones? This bears on the various figures quantifying pause occurrences in Figure 3 (which count both seen and unseen pauses). Second, for Figure 2/2C, what does net velocity with pauses vs. force look like? It will be good to give the readers a sense of the sensitivity to pause-removal of the force/velocity dependence.

The reviewers are correct in pointing out the limitations of pause detection due to both time resolution and experimental noise. However, the issue of experimental noise is in large part mitigated by the fast translocation velocities of SpoIIIE. Due to the large changes in DNA contour length resulting from SpoIIIE translocation, even pauses as short as 30-50 msec can be easily identified (the faster the motor translocates, the shorter pauses we can detect). To this end, we have included a supplemental figure showing several detected pauses at different ATP concentrations. We also highlighted pauses in Figure 3.

As for how we handle pause analysis – we follow the same approach as described in our recent study to which the current paper represents a research advance, so in the interest of brevity we didn’t duplicate the Data Analysis and Materials and methods section from Liu et al., 2015, instead we made a brief comment and cited the previous study: “The number and duration of pauses missed by this algorithm were inferred by fitting the pause durations to a single exponential with a maximum likelihood estimator (Liu et al., 2015).”

The reviewers asked “How many such "unseen" pauses are there relative to the visible ones?” – the answer depends on the mean pause lifetime for a particular distribution and the shortest pauses that we can detect. As shown in Figure 3 inset, the distribution of detectable pause durations at 0.25mM ATP is very well described by a single exponential distribution with a mean pause lifetime of ~28ms. We assumed that the short pauses that go undetected follow the same distribution. For the 0.25mM ATP dataset shown in Figure 3 inset, mean pause lifetime = 28ms, shortest detectable pause = 50ms, we are detecting an estimated ~17% of all the pauses, whereas the remaining ~83% of the predicted pauses are shorter than 50 ms and go undetected.

As mentioned above, the duration of the shortest detectable pause depends on the SpoIIIE translocation velocity and the signal to noise in the experiment. For 0.25 mM and 0.50 mM ATP, we could reliably detect pauses of 50 ms or longer, whereas at higher [ATP] we could identify pauses as short at 30 ms – similar to the thresholds used in our previous study (Liu et al., 2015).

We have revised Figure 3 to show both “measured pause density” (referring to pauses we actually detect) and “corrected pause density” (which accounts for the missed pauses). We have also updated the text and the figure captions referring to Figure 3. Note, that the contribution of pauses missed to the dynamics of translocation of the enzyme gets smaller and smaller the shorter they become. In other words, although we cannot detect short-lived pauses which comprise most of the pausing events in the SpoIIIE data, when the number missed pauses is accounted for, the overall trends displayed in Figure 3 remain the same, as do the resulting conclusions. Note that we did not correct for missed pauses in Figure 3 as it is intended to show that pause density does not depend on force and for a given curve the correction factor would be the same across all forces.

The reviewers asked “Second, for Figure 2/2C, what does net velocity with pauses vs. force look like? It will be good to give the readers a sense of the sensitivity to pause-removal of the force/velocity dependence.”

Thank you for raising this point. We have previously done this analysis but initially chose not to include it in the manuscript given the brief nature of the “research advance” article format. We generated Figure 2—figure supplement 4 showing plots of the pause-free velocity (i.e. velocity after the detectable pauses were removed) and the net velocity (i.e. no pauses were removed). We find that the net translocation velocity is only 10-20% lower than the pause-free translocation velocity. We conclude that pause removal has a minor effect on velocity and does not change the overall trends in force-velocity dependence or our conclusions derived from it.

3) The main-text statement (subsection “The SpoIIIE Cycle Contains at Least Two Force-Dependent Kinetic Rates”) that the velocity "rapidly decreases again beyond 40 pN" in saturating ATP conditions seems to be contradicted by the data in Figure 2, where the sharp decrease seems only to be clear in the 3 mM data. The other ATP concentrations show instead a broad, continuous decrease in velocity rather than a clear high-force knee. Further, there is clearly large error in the high-force velocity points, as evidenced in Figure 2. This reduces confidence in the analysis based on the combination of the 2, 3 and 5 mM ATP data in Figure 2. On that note-- how are those data points even combined, given the different force values used for each [ATP] data set (as seen in Figure 2)?

The reviewers rightfully pointed out that the velocity drop off at high forces is more pronounced at some ATP concentrations. The reviewers also correctly point out that velocity measurements at high forces have large error-bars associated with them. Both of these features are a result of the amount of data that we were able to collect at various forces (see Table 1). For example, individual velocity measurements at a low force (5-10pN) were made using 30-60 kb of translocation data per measurement resulting in small error-bars (standard error of the mean) of only 40-100 bp/sec (for velocities of 3000-4000 bp/sec). In contrast, velocity measurements at very high forces (40-50kb) were made using as little as 1-2kb of translocation data per measurement resulting in error-bars of 200-400bp/sec.

Although we attempted to collect as much data as practically possible, we had only minimal coverage at very high forces. As a result, some force velocity curves display a gentle/broad decrease in velocity whereas others display a more sharp decrease in velocity at high forces (Figure 2, black, blue, and green curves). In our opinion these inconsistencies are likely due to the large error-bars associated with velocity measurements at 40-50pN. We reasoned that the force-velocity behavior of SpoIIIE at near-saturating [ATP] of 5, 3 and 2 mM would largely be unchanged, as evidenced by the similar translocation velocities of SpoIIIE in the low force (<30 pN) regime. Since the error-bars of these data sets overlap in the high-force regime, we also reasoned that combining the force-velocity data at 5, 3, and 2mM [ATP] would still faithfully capture SpoIIIE’s force-velocity behavior above 30-40 pN.

The data for velocity vs. force measurements was collected in passive mode where the opposing force increases gradually as the motor translocates DNA. The individual translocation traces were segmented in 2pN windows and the translocation velocity was computed for each force window. To combine 2mM, 3mM, and 5mM datasets we pooled the velocity measurements from individual force windows and then binned them.

We would like to point out that the reduction in velocity at high forces is also noticeable in the ADP titration data (Figure 2) and Phosphate titration data (Figure 2) although those datasets also have limited coverage at high forces (and therefore relatively large error-bars). Ultimately the velocity of any molecular motor is bound to decrease at very high opposing force, the question is how steeply (or gently) does the SpoIIIE velocity drop off with increasing force. That will affect the value of the distance to the transition state derived from the linear model or the branched model (Figure 2). We can still confidently state that this force-velocity curve cannot be described by a model with a single force-sensitive transition, but we cannot accurately fit the velocity drop off at high forces. We want to emphasize that this conclusion is independent of how steep or gentle the drop off is.

To address these comments, we edited this section of the manuscript to explain the rationale behind combining a few datasets into a single force-velocity curve, and we edited the following sentence (“rapidly” is subjective):

“At near-saturating [ATP], SpoIIIE exhibits a bi-phasic force-velocity dependence the pause-free velocity drops between 5 and 15 pN, remains relatively force-insensitive between 15 and 40 pN, then decreases again beyond 40 pN (Figure 2).”